# Fatigue Crack Propagation under Christmas Tree Load Pattern

Diogo M. Neto *, Edmundo R. Sérgio, Francisco Jesus, James C. Newman, Jr. and Fernando V. Antunes

Centre for Mechanical Engineering, Materials and Processes (CEMMPRE), Department of Mechanical Engineering, Univ Coimbra, 3030-788 Coimbra, Portugal
* Correspondence: diogo.neto@dem.uc.pt

**Abstract:** Most mechanical components are subject to dynamic loads, which can cause failure in service. This study aims to evaluate the effect of variable amplitude loadings on fatigue crack growth (FCG) in CT specimens produced with the AA2024-T351 aluminum alloy. Specifically, it is intended to predict the FCG rate when the specimen is subjected to a complex loading pattern, named the "Christmas Tree Spectrum". Crack growth is simulated by node release, which occurs when the cumulative plastic strain at the crack tip reaches a critical value (110%) that is supposed to be a material property. It is therefore assumed that cyclic plastic deformation is the main damage mechanism. The specimen was subjected to four different load patterns: the 6–60 N low-frequency constant amplitude load (CAL); the 6–21 N high-frequency CAL; the Christmas Tree (15–9) and the Christmas Tree (9–3) patterns. The Christmas Tree 15–9 load pattern is defined by nine increments of +15 N and −9 N followed by eight increments of +9 N and −15 N. The results indicate that the Christmas Tree (15–9) pattern increases crack tip damage relative to the constant amplitude loading. This is attributed to small variations in material hardening, particularly during the unloading phase of the load block. On the other hand, the Christmas Tree (9–3) pattern did not show a significant effect, indicating the importance of the range of small-amplitude cycles. The crack closure phenomenon is usually used explain the effect of loading parameters, but this is an exception.

**Keywords:** fatigue crack growth; 2024-T351 aluminum alloy; Christmas Tree spectrum; cumulative plastic strain; material hardening; crack closure

## 1. Introduction

Components and structures are typically submitted to cyclic loads and therefore must be designed against fatigue. In the damage tolerance approach, initial cracks are assumed to exist, with a size equal to the limit of detection techniques. The time between inspections is defined from the number of load cycles required to propagate the crack up to a critical length. The accuracy of the fatigue crack growth (FCG) rate is therefore crucial in this design approach.

The FCG is a complex phenomenon, which is affected by different parameters, namely, material, geometry, environment and loading conditions. The effect of loading parameters, in particular, has been widely studied but is not completely understood. In practice, the loads applied to components and structures are usually complex, with variable frequency and amplitude. Standard load patterns were defined, for different practical applications, which facilitate the comparison of results [1]. The CARLOS load pattern was defined for automotive applications, WISPER was the loading sequence suggested for wind turbines, while TWIST and FALSTAFF sequences were proposed for transport and fighter aircraft [2]. The TWIST load spectrum, for example, has an average of a thousand small gust cycles per flight with a major cycle associated with the ground–air–ground sequence and a few overloads associated with more severe atmospheric turbulence.

However, to understand the mechanisms that explain the effect of these complex loading patterns, it is recommended to follow a strategy of increasing complexity. Accordingly, the most studied loading pattern is the constant amplitude loading with different stress

ratios. A crack closure phenomenon was proposed to be the main mechanism explaining the effect of stress ratio [3]. However, other researchers claim that the contact of crack flanks is not relevant and that the FCG rate is dependent on the maximum load and load range [4], while the residual stresses ahead of the crack tip play a major role [5]. The fundamental mechanisms of variable amplitude loading have been studied by applying overloads, underloads and load blocks. The main mechanism is proposed to be plasticity-induced crack closure [6–8], but other mechanisms have been used to explain the trends observed, namely crack tip blunting [9], residual stresses [10,11], strain hardening [12,13] and crack branching [14]. In previous numerical works of the authors [8,15], the transient effects after overloads and load blocks were explained by plasticity-induced crack closure, which depends on residual plastic wake and crack tip blunting. In fact, the construction of numerical models without contact between crack flanks showed that the transient regimes of da/dN disappear; therefore, these are a result of the crack closure phenomenon.

In this strategy of increasing complexity, approaching realistic load patterns, a novel loading pattern, called Superblock 2020, was proposed by Sunder [16]. It is composed of eight load blocks, with the same maximum load, separated by overloads. The experimental study of Sunder [16] illustrated a systematic cycle-sequence sensitivity of FCG, the magnitude of which increased with the decreasing amplitude of baseline cycles. These trends were attributed to the effect of near-tip residual stress on the intrinsic threshold stress intensity. However, the numerical study of Neto et al. [17] indicated that plasticity-induced crack closure, once again, was the main mechanism behind the trends observed. A load pattern called Christmas Tree was designed by James Newman [18] to test the Rainflow-on-the-Fly option in FASTRAN [19]. However, the transient effects associated with this pattern are not fully understood.

Therefore, the objective here is to develop a numerical study of the Christmas Tree load pattern in order to understand the fundamental mechanisms. The numerical model assumes that cyclic plastic deformation is the main damage responsible for FCG and that the crack-driving parameter is the cumulative plastic strain at the crack tip. This approach was validated in previous works by the direct comparison with experimental results. Borges et al. [20] successfully predicted the effect of $\Delta K$ observed experimentally in AA2024-T251 and 18Ni300 steel, Neto et al. [8] predicted the effect of stress ratio, Neto et al. [17] predicted the effect of Superblock 2020 load pattern and Neto et al. [15] predicted the effect of load blocks.

## 2. Numerical Model

The numerical analysis of FCG under variable amplitude loading was carried out using the in-house finite element code DD3IMP, which was originally developed to simulate sheet-metal-forming processes [21]. Although the plastic zone size was effectively much smaller than the elastic region of the specimen, the mechanical behavior of the specimen was assumed elastic–plastic. Indeed, the crack tip was always surrounded by a plastic zone, dictating the crack propagation (irreversible phenomenon). Therefore, the adopted numerical approach assumed that cyclic plastic deformation is the main damage mechanism responsible for FCG and uses the cumulative plastic strain at the crack tip to define the node release [20]. This criterion states that when a critical value of accumulated plastic deformation is reached, the node containing the crack tip is released, i.e., the propagation occurs. The critical value of the accumulated plastic deformation for the AA2024-T351 alloy was previously calibrated by comparing experimental and numerical values of the FCG rate under constant amplitude loading [22]. The critical value of accumulated plastic deformation found was 1.10, which is considered a material property.

### 2.1. Specimen

All numerical tests were performed using a CT specimen, presenting the initial crack length, $a_0$, of 15 mm and width, $W$, of 36 mm. The initial position of the crack tip is indicated by an arrow. Since the CT specimen presents two symmetric planes, only 1/4 of

the geometry was modeled. In addition, the numerical analyses were performed under a plane stress state by reducing the thickness of the specimen to 0.1 mm and allowing out-of-plane displacements. The finite element mesh used in the numerical simulations is shown in Figure 1, which was composed of 12,590 finite elements and 25,676 nodes. The model employed linear hexahedral finite elements, where the size of the finite elements was gradually reduced to achieve a refined mesh in the vicinity of the crack path. Indeed, the mesh size near the crack tip was 8 μm, allowing one to accurately predict local strain and stress gradients. In addition, only one layer of finite elements was used along the thickness direction. The approach followed to predict da/dN was not significantly affected by the size of crack tip elements [23]. In fact, FCG was calculated using da/dN = Δa/ΔN. Each crack increment, Δ*a*, corresponds to one finite element, while ΔN is the number of load cycles needed to reach the cumulative plastic strain. A decrease in mesh size to 4 μm, for example, would increase the stress and strain levels at the crack tip, therefore reducing ΔN. So, a decrease in mesh size is compensated for by the decrease in ΔN, and there is significant robustness in the numerical procedure relative to mesh size.

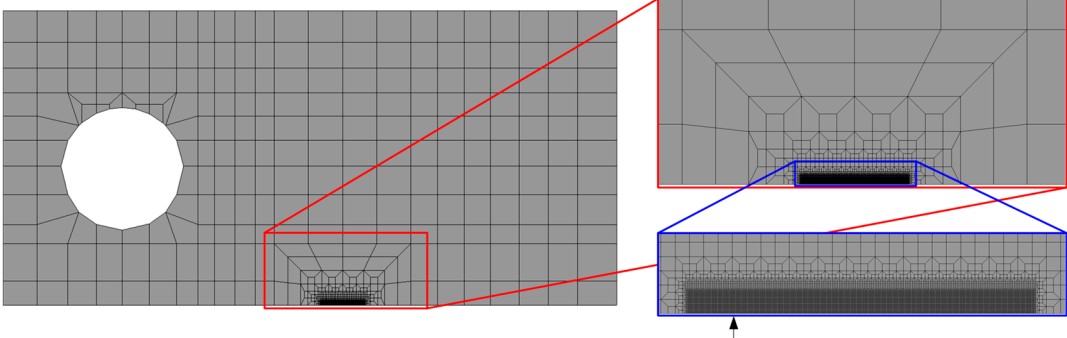

**Figure 1.** Finite element mesh of the specimen used in the numerical simulation.

Since only a part of the specimen was modeled, the boundary conditions were properly defined, i.e., adequate boundary conditions at the nodes located in the horizontal symmetry plane were applied to define the initial crack length. The physical contact between the crack flanks during the unloading stage is essential for the accurate modeling of crack closure. Thus, the contact between the flanks of the crack was modeled considering a rigid plane surface (frictionless) aligned with the crack symmetry plane.

*2.2. Material*

The material used in this study was 2024-T351 aluminum alloy. The elastic behavior was defined by Hooke's law using the Young's modulus (*E*) and Poisson ratio (*ν*), both listed in Table 1. The plastic behavior was described with the von Mises yield criterion coupled with a mixed hardening model using the Swift isotropic and Lemaitre–Chaboche kinematic hardening laws, under an associated flow rule. The Swift isotropic hardening law [24] is given by:

**Table 1.** Elastic properties and material parameters for the 2024-T351 aluminum alloy. Adapted from [22].

| *E* (GPa) | *ν* | $Y_0$ (MPa) | *K* (MPa) | *n* | $X_{Sat}$ (MPa) | $C_X$ |
|---|---|---|---|---|---|---|
| 72.26 | 0.29 | 288.96 | 389.0 | 0.056 | 111.84 | 138.80 |

$$Y = C\left[\left(\frac{Y_0}{C}\right)^{\frac{1}{n}} + \bar{\varepsilon}^p\right]^n,\tag{1}$$

where $Y$ is the flow stress and $\bar{\varepsilon}^p$ denotes the equivalent plastic strain. The material parameters of Swift law are $Y_0$, $C$ and $n$. The Lemaître–Chaboche law describes non-linear kinematic hardening as follows [25]:

$$\dot{X} = C_x \left[ X_{Sat} \frac{\sigma\prime - X}{\bar{\sigma}} - X \right] \dot{\bar{\varepsilon}}^p,$$ (2)

where $\dot{X}$ is the back-stress tensor, $\bar{\sigma}$ is the equivalent stress and $\dot{\bar{\varepsilon}}^p$ is the equivalent plastic strain rate. $C_X$ and $X_{Sat}$ are the material parameters of the Lemaître–Chaboche law, which denote the saturation rate and the norm of the saturated back-stress tensor, respectively. The calibration of the material parameters was performed by minimizing the difference between the numerical and the experimental results of stress–strain curves obtained in low-cycle fatigue tests. The obtained set of material parameters is presented in Table 1.

### 2.3. Load Patterns

Four different load patterns were applied to the specimen, covering both constant and variable amplitude loading, as shown in Figure 2. Both the Christmas Tree (15–9) and the Christmas Tree (9–3) patterns are variable-amplitude loading patterns, where high cyclic loading is embedded with many small cycles, ranging from 6 N to 60 N. The load pattern Christmas Tree (15–9) is defined by increments of +15 N and −9 N during the global increase in the load. During the global decrease in the load, the pattern contains increments of −15 N and +9 N. On the other hand, the load pattern Christmas Tree (9–3) is defined by increments of +9 N and −3 N during the global increase in the load. During the global decrease in the load, the pattern contains increments of −9 N and +3 N. The difference between these two loadings is that Christmas Tree (15–9) has steeper unloads (see Figure 2). In addition to the variable-amplitude loading patterns, two different load patterns with constant amplitude were also evaluated. The low-frequency loading pattern ranges from 6 N to 60 N, which are the limits of both Christmas Tree patterns, as shown in Figure 2. The load in the high-frequency pattern ranges from 6 N to 21 N, which represents the maximum load range of each load cycle of the Christmas Tree (15–9) load pattern.

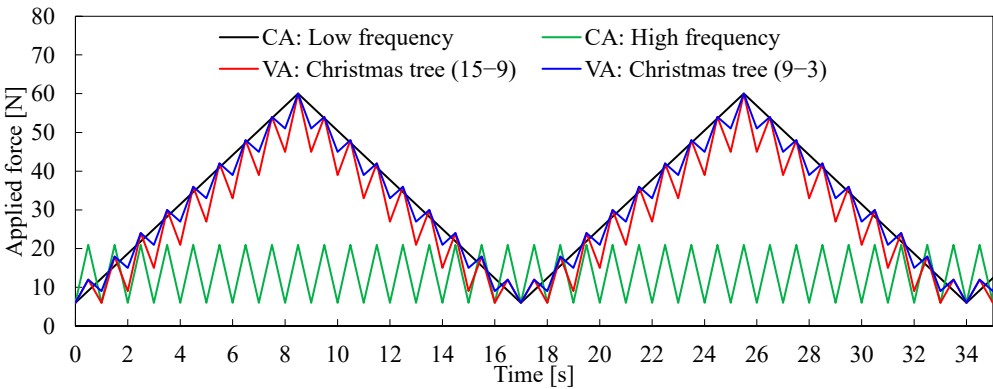

**Figure 2.** Four different load patterns used in the numerical simulation (CA: constant amplitude; VA: variable amplitude).

Both the Christmas Tree patterns and the low-frequency loading pattern require 17 s to define the pattern (load block), while the high-frequency pattern only requires 1 s (see Figure 2). Nevertheless, considering each load cycle applied in the patterns, the load frequency is $f$ = 1 Hz for all load patterns, except for the low-frequency loading pattern which has a load frequency of $f$ = 0.058 Hz. Considering constant-amplitude loading conditions, the FCG rate is usually defined by the calculation of crack growth per cycle. Nevertheless, a time-based formulation is more suitable for general variable random loading [26]. The basic concept of the time-based FCG rate is $da/dt = (da/dN)\,f$, with $a$ being the crack length and $N$ the number of load cycles.

## 3. Results

### 3.1. FCG Rate

Figure 3 presents the evolution of the predicted FCG rate for each load pattern studied. Both Christmas Tree loading spectrums provide identical FCG rate evolutions, which are coincident with the trend obtained for the low-frequency pattern. Therefore, the effect of the small cycles embedded into the high cyclic loading seems unimportant. Considering the variable-amplitude loading spectrums, the predicted FCG rate increases at the beginning (until about 15.08 mm of crack length). This increase is linked with the stabilization of cyclic plastic deformation. Then, the FCG rate decreases gradually, where the crack closure level is increasing due to the formation of the residual plastic wake, until a steady-state regime is achieved (about 0.19 µm/s). The high-frequency pattern does not exhibit the same initial peak, stabilizing very quickly at about 0.07 µm/s, i.e., about 3 times smaller than the other loading spectrums. Note that the number of load cycles required to achieve a crack length of 15.30 mm for both Christmas Tree patterns is about 1400 cycles, while the low-frequency pattern only requires around 90 load cycles.

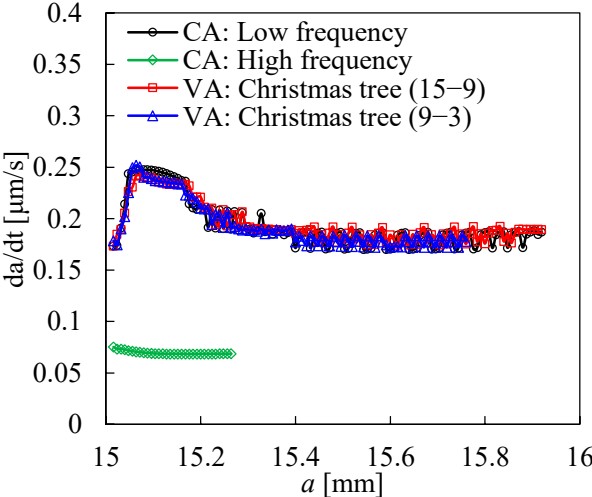

**Figure 3.** Evolution of the predicted FCG rate for the different loading spectrums.

### 3.2. CTOD Plots

The analysis of Crack Tip Open Displacement (CTOD) plots to understand what is happening at the crack tip is very interesting. Therefore, for each loading spectrum presented in Figure 2, the CTOD curve was evaluated at the node immediately behind the crack tip (i.e., 8 µm behind) for the crack length $a$ = 15.30 mm. Figure 4a presents the evolution of the CTOD, comparing the four load patterns. The low-frequency pattern produces an open loop, with similar values of elastic and plastic CTOD, indicated by $\Delta\delta_e$ and $\Delta\delta_p$, respectively. The minimum load is 6 N, but the crack only opens above 20 N due to plasticity-induced crack closure.

The CTOD curves of the Christmas Tree patterns contain a kind of knurling due to the small cycles embedded into the high cyclic loading. Since each unloading stage of the Christmas Tree (15–9) pattern is steeper in comparison with the Christmas Tree (9–3) pattern, the knurling on the CTOD curves is also steeper. All the small cycles are linear, without any signal of plastic deformation, and the slope of all of them is the same, corresponding to the elastic behavior of a crack with $a$ = 15.30 mm. Globally, the CTOD predicted for both Christmas Tree loading spectrums is similar to the one obtained for the low-frequency pattern, which agrees with the predicted FCG rate (see Figure 3). There is only a small downward movement of the loop with the imposition of the small cycles, which means that the crack opening load is higher in the Christmas Tree (15–9) pattern.

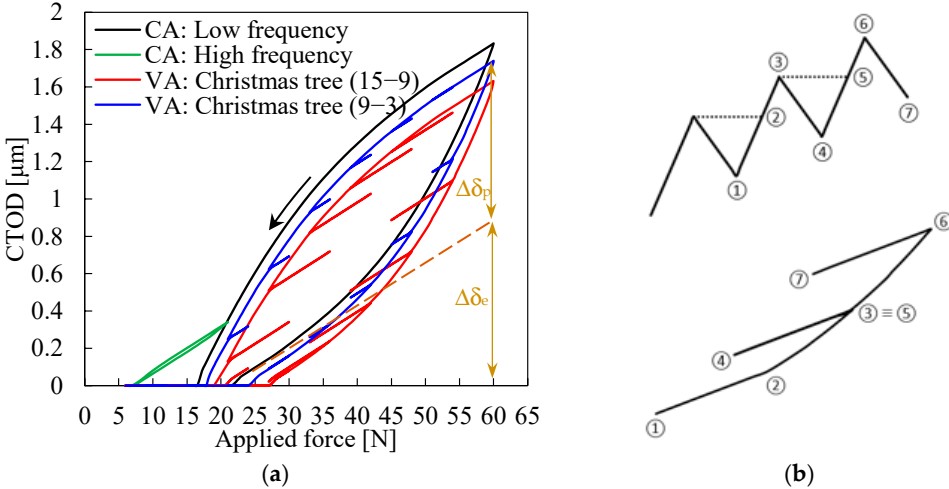

**Figure 4.** Comparison of different loading spectrums: (**a**) CTOD curves evaluated for *a* = 15.3 mm. (**b**) Detail of applied load and of CTOD versus load plot.

On the other hand, in the case of the high-frequency loading pattern, the elastic CTOD is about 16 times higher than the plastic CTOD ($\Delta\delta_e$ = 0.64 μm and $\Delta\delta_p$ = 0.04 μm). This explains the nearly closed CTOD loop. This small value of plastic CTOD, i.e., of small crack tip plastic deformation, leads to a lower value of the predicted FCG rate, as shown in Figure 3.

Figure 4b details the variation in CTOD with applied force in the case of Christmas Tree load patterns. The increase in load from the local minimum (point 1) produces elastic deformation up to point 2, which corresponds to the previous local maximum load. Above point 2, the material starts suffering from plastic deformation, which increases up to the new local maximum load (point 3). The subsequent unloading (3–4) only produces elastic deformation. The plastic deformation occurs in the loading and starts after point 5, increasing up to the new local maximum (6).

*3.3. Plastic Strain*

Figure 5a presents the evolution of the plastic strain at the node containing the crack tip during some node releases starting at a crack length *a* = 15.30 mm. The plastic strain at the beginning of each new propagation is approximately 0.8 for all loading patterns. This results from the accumulation of damage while the node is positioned some distance ahead of the crack tip. On the other hand, for the high-frequency loading pattern, the deformation starts almost from zero. Therefore, the size of the plastic zone associated with the high-frequency loading pattern is very small in comparison to the other loading patterns studied. Due to the relatively low load range (see Figure 2), only one element ahead the tip is accumulating plastic deformation. Thus, this loading pattern requires more load cycles to reach the critical value of accumulated plastic deformation, yielding a lower FCG rate than the other patterns. The horizontal dashed line defines the critical value of cumulative plastic strain. Node release occurs when the accumulated strain value exceeds this limit. Despite the difference in the loading patterns (see Figure 2), the overall plastic strain growth rate between nodal releases is identical for all load spectrums, as shown in Figure 5a.

Figure 5b is a detail of the evolution of the plastic strain evaluated at the crack tip for *a* = 15.30 mm for the different load patterns. In order to compare the behavior of the increase in plastic deformation during the same time interval, 17 load cycles of the high-frequency pattern were used. For the remaining load patterns, only one load block is shown in Figure 5b. Note that the plastic deformation graphs have undergone a translation of both axes, setting both the initial plastic strain and time to zero. Considering the constant-amplitude loading cases, both present exponential growths in plastic deformation

at the crack tip. However, the low-frequency pattern presents more prominent increases, in comparison to the high-frequency one, due to the higher $\Delta K$ levels. For both Christmas Tree loading patterns, the plastic deformation has several increasing steps. These steps occur due to successive loading and unloading phases. During the global loading phase (0–8.5 s), the magnitude of these jumps increases, following the exponential growth observed for the constant amplitude case. A similar trend is observed during the global unloading phase (8.5–17 s). The biggest "jump" in the plastic strain occurs at the maximum load, for a time of 8.5 s. Additionally, the Christmas Tree (15–9) curve is above the results obtained for the CA low frequency and for the 9–3 Christmas Tree, which is particularly evident in the global unloading phase. This is relevant, as it indicates an influence of small cycles on crack tip damage quantified by cumulative plastic strain. This influence is not evident in the da/dN results presented in Figure 3 due to the oscillatory behavior associated with the discrete character of the numerical procedure. In addition, this does not occur for the 9–3 Christmas Tree load pattern, since the corresponding curve overlaps the CA curve in Figure 5b.

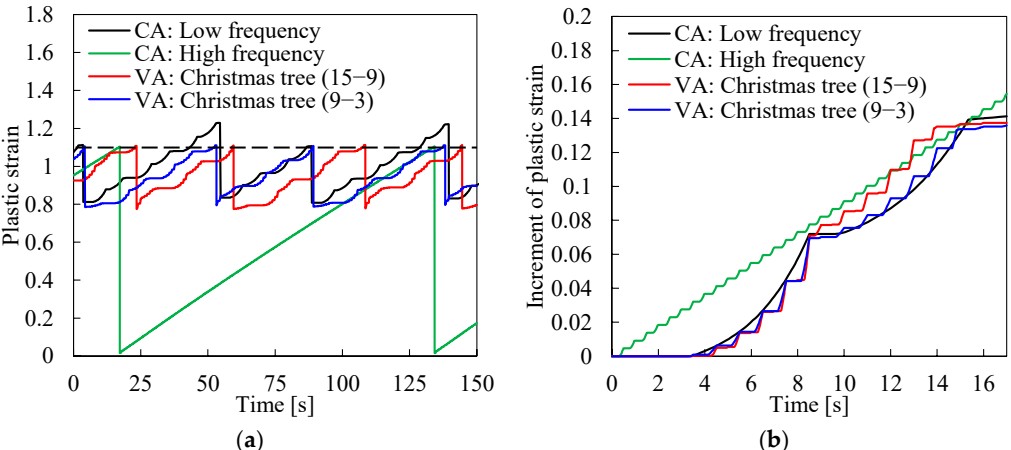

**Figure 5.** Comparison of different loading spectrums: (**a**) plastic strain evaluated at the node containing the crack tip; (**b**) increment in plastic strain evaluated at the node containing the crack tip.

In order to better understand the effect of the loading patterns on the plastic deformation at the crack tip, the portion of the loading cycle with plastic strain rise was identified. The results are presented, in Figure 6, for the constant-amplitude loading patterns.

Considering the low-frequency loading pattern (Figure 6a,c), before the crack opening loading, $F_{open}$ = 21.55 N, there is no plastic deformation. After its opening, it enters in the elastic domain for about half a second, where there is still no plastic deformation. After around 3 s from the beginning of the load cycle, the plastic deformation starts and increases exponentially until it reaches the maximum load. When the unloading starts, the crack recovers the elastic behavior and, once again, enters the plastic domain where the plastic deformation increases, also exponentially, until the crack closes at $F_{closure}$ = 16.73 N. After the crack closure, indicated by the vertical dashed line, plastic deformation still increases, but with a slight linear rise. This is important because it indicates that there is damage after the crack closure, and therefore, the concept of effective load range is not totally correct.

In the high-frequency load pattern (Figure 6b,d), the plastic deformation also increases exponentially. After the crack opening, at $F_{open}$ = 7.08 N, the crack is in the elastic domain and only enters the plastic domain when the loading reaches values close to the peak. When the loading starts to decrease, the crack behaves elastically, as it does with the low-frequency pattern. The crack again has an increase in plastic deformation, also in an exponential way, when the applied force values reach low values. After the crack closure occurs at $F_{closure}$ = 6.88 N, the plastic deformation continues to increase, but in a less pronounced way. Note that the CTOD of this loading pattern (see Figure 4a) is predominantly elastic, making the plastic deformation, in this loading, very small. The increase in plastic strain at

the crack tip in a single load cycle under the high-frequency load pattern is about 15 times smaller than in the low-frequency load pattern (compare Figure 6c,d).

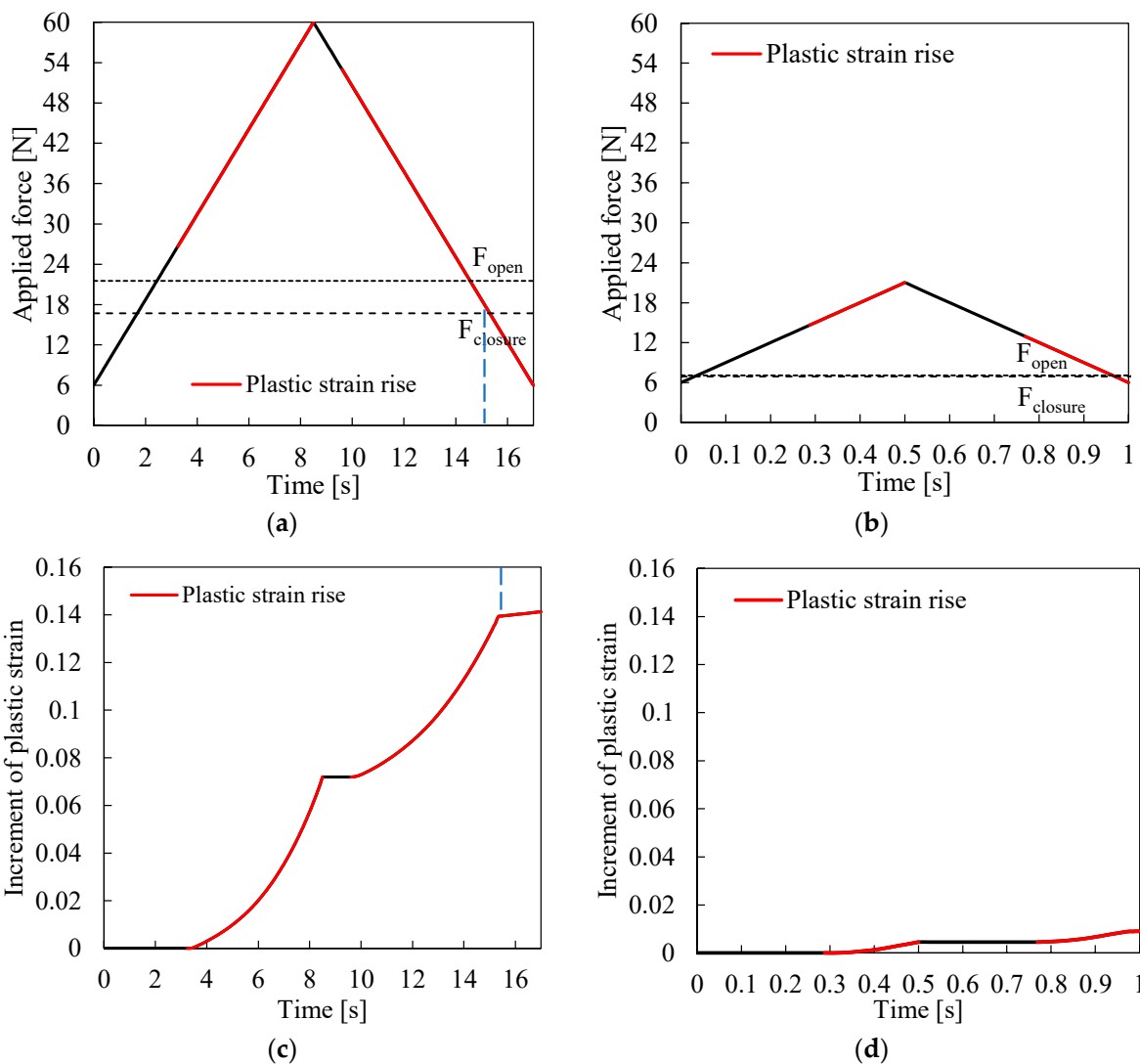

**Figure 6.** Evolution of plastic deformation during loading to a = 15.30 mm: (**a**) low-frequency pattern; (**b**) plastic strain rise of low-frequency pattern; (**c**) low-frequency pattern; (**d**) plastic strain rise of high-frequency pattern.

Figure 7a,b show the portion of the Christmas Tree (15–9) and Christmas Tree (9–3) patterns, respectively, where an increase in plastic deformation occurs. This is indicated by the red segments. The increase in plastic deformation before the crack opening is negligible for both load patterns. The crack opening load is $F_{open}$ = 27.11 N for the Christmas Tree (15–9) pattern and $F_{open}$ = 24 N for the Christmas Tree (9–3) pattern. Until the loading reaches an overall maximum peak of the load block, plastic deformation occurs essentially at the local load peaks. As analyzed in Figure 4b, plastic deformation begins when the load exceeds the previous local maximum. There is also some deformation during unloading, close to the local minimum loads, but only for the Christmas Tree (15–9) pattern. When the loading starts to have global decreasing behavior, after 8.5 s, the plastic deformation starts to manifest itself in the valleys of successive unloading cycles. Reversed deformation begins when the load drops below the previous local minimum. There is also some deformation during loading, close the local maximum values, but only for the Christmas Tree (15–9) pattern.

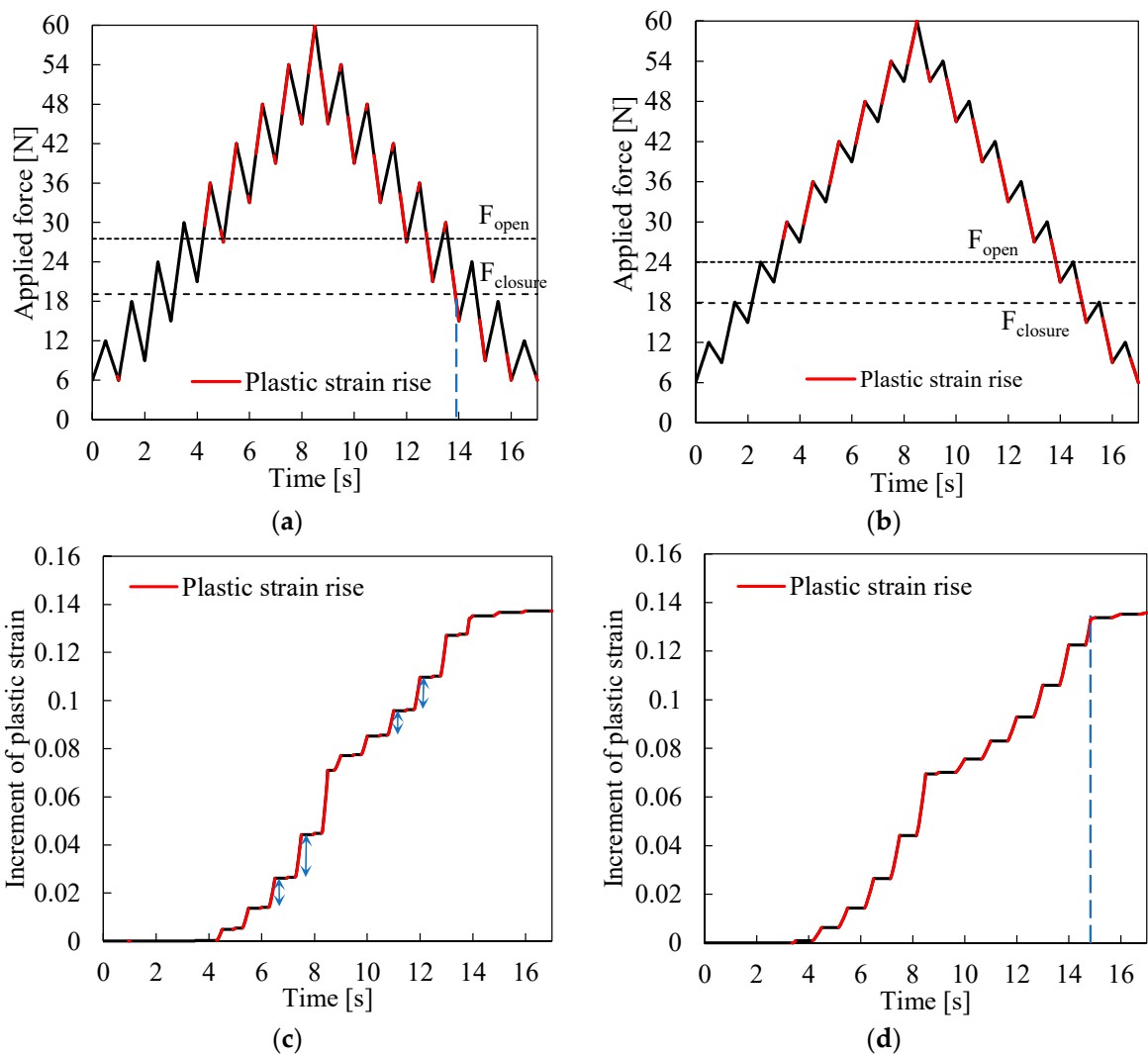

**Figure 7.** Evolution of plastic deformation during loading to a = 15.30 mm: (**a**) Christmas Tree (15–9) pattern; (**b**) Christmas Tree (9–3) pattern; (**c**) plastic strain rise of Christmas Tree (15–9) pattern; (**d**) plastic strain rise of Christmas Tree (9–3) pattern.

Figure 7c,d show the increase in plastic strain for the load blocks presented in Figure 7a,b, respectively. In the global loading phase, as indicated by the arrows, there is a progressive increase in plastic strain variation, which is according to the exponential increase observed in Figure 6c. A similar trend is observed during the unloading phase. Note also that plastic strain keeps on increasing, even after crack closes (as indicated by the vertical dashed line in Figure 7a,c), at $F_{closure}$ = 19.13 N and $F_{closure}$ = 17.85 N for the Christmas Tree (15–9) and Christmas Tree (9–3) patterns, respectively. Nevertheless, after closure, the increase in plastic deformation becomes much less pronounced than when the crack is open. This, once again, destroys the purity of the effective load range, which usually represents the crack closure phenomenon.

### 3.4. Without Contact of Crack Flanks

The numerical elimination of the contact between the crack flanks is interesting to understand the relevance of the crack closure phenomenon. Figure 8a shows the FCG rates for the four load patterns neglecting the contact between the crack flanks. The da/dt curves of these load patterns behave similarly to those obtained with contact (see Figure 3). However, non-contact loading patterns show a more pronounced initial increase, and their stabilization value tends to increase linearly along the crack length. Note that, although the

value of da/dt also increases linearly, along the crack length in the simulations with contact, this is less evident, when compared with non-contact results. Indeed, da/dt increases because during propagation, the crack length gets larger and larger, which implies a growth in the magnitude of crack tip fields, and so of the stress intensity factor and propagation rate. Finally, here, the overall values of da/dt are superior to those obtained in the simulations with contact, where the stabilization value is 0.33 μm/s, as the protective effect provided by crack closure is not present. Similarly to Figure 3, the high-frequency loading is also an exception, presenting a da/dt value of about 0.08 μm/s; this is around 4 times lower than the other load patterns. The oscillations that the low-frequency pattern present are caused by the relatively high value of ΔK. The propagation always occurs at the minimum value of the load cycle; therefore, there may exist some oscillation in the number of load cycles required to reach the critical cumulative plastic strain. On the other hand, for the VA loading patterns, although the ΔK is similar, the crack propagation on these patterns can occur for different values of force (any local minimum), which tend to reduce the oscillations in a node release strategy.

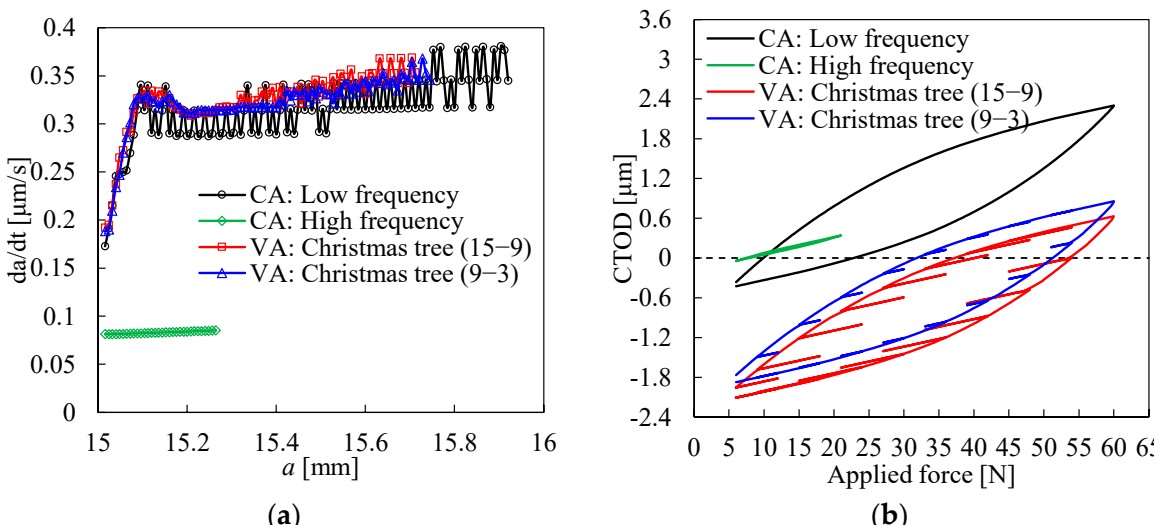

(**a**)                        (**b**)

**Figure 8.** Comparison of different loading spectrums neglecting the contact between the crack flanks: (**a**) da/dt versus crack length; (**b**) CTOD curves for *a* = 15.30 mm.

Figure 8b presents the CTOD plots obtained without contact between the crack flanks. Negative values are obtained, which indicate an overlap of the crack flanks at the minimum load, which cannot occur in a physical specimen. The crack tends to open immediately from the minimum load, unlike what happens in Figure 4a, in which the contact between the crack flanks is taken into account. Anyway, the shapes of the CTOD curves are identical to the curves obtained with contact. Note that the CTOD curves of the variable amplitude loadings were obtained after a node release at very low loading. Since the crack propagation did not occur at the global minimum load of the load block, a translation of the CTOD curves of the VA loading patterns occurred.

Figure 9 shows the evolution of the plastic deformation evaluated at the crack tip for *a* = 15.30 mm, comparing different load patterns from the non-contact simulations. The plastic strain behavior of each simulation has several similarities with the results obtained considering the contact of the crack flanks (Figure 5b). Compared with the situation of constant-amplitude loading, the effect of the Christmas Tree (15–9) loading case is evident, particularly in the unloading part of the loading block, being more evident than that observed in Figure 5b for the situation with contact. The Christmas Tree (9–3) loading case does not produce an increase in damage, once again.

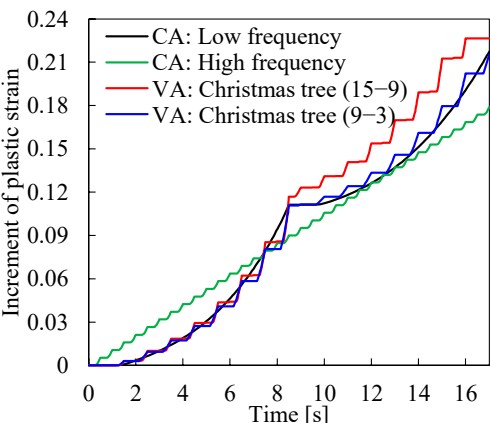

**Figure 9.** Comparison of plastic strain curves for point *a* = 15.30 mm in non-contact simulations.

## 4. Discussion

In the Christmas Tree spectrum, high cyclic loading is embedded with many small cycles. The da/dN results indicate that there is no significant influence of the short-range cycles on the overall FCG rate, but there is some oscillation of the numerical predictions. However, a closer look to the accumulation of plastic strain indicates an increase in the FCG rate for the Christmas Tree (15–9) load pattern, particularly during the unloading phase of the load block. This is according to the experimental results of Newman [26], who observed the influence of small-amplitude cycles on a-N curves. The increase in $\Delta P/P_{max}$ from 5% to 20% increased the crack propagation rate, i.e., the small-amplitude cycles have a deleterious effect on FCG rate. This difference was evident only for a large number of load blocks, above 30,000. The small ranges considered here were $\Delta P/P_{max}$ from 5% (=3/60) and 15% (=9/60), and therefore are of the same order of those considered by Newman [27]. Note that the materials studied are different (Newman studied 7075-T7351 aluminum alloy, while the material studied here was 2024-T351 aluminum alloy), which did not prevent similar trends from being obtained.

The mechanism behind the effect of the Christmas Tree load pattern is material hardening. The small-range cycles slightly modify the complex material hardening, particularly during the unloading part of the load block. This effect is even more evident when the contact of crack flanks is numerically eliminated. The effect of loading parameters is usually linked to the crack closure phenomenon, but this is not the case. In fact, crack closure was able to explain the effect of overloads [8], load blocks [15] and the Superblock2020 load pattern [17]. Therefore, the dominance of hardening effects is an exception in this context.

Additionally, here, it is assumed that cyclic plastic deformation is the crack tip damage mechanism responsible for FCG. However, other mechanisms may be active, namely the brittle mechanism or the growth and coalescence of microvoids, which may increase the effect of the small-range cycles. On the other hand, the small-range cycles may even have a protective effect. In fact, these load cycles promote the formation of oxide debris [28–30], leading to oxide-induced crack closure (OICC). This means that even though the application of small-load amplitudes does not contribute to crack growth, the subsequent crack growth produced by load cycles above the threshold is affected by the developed oxide debris.

Finally, according to Newman [27], crack-growth damage only occurs during loading. The unloading part of the small-amplitude cycles causes reverse plastic deformations which may affect damage during the next load application. The present numerical results, based on cumulative plastic strain, clearly indicate that damage occurs during both loading and unloading.

## 5. Conclusions

In this paper, a numerical study on the effect of variable-amplitude loadings on FCG was carried out. The so-called Christmas Tree loading pattern was compared with constant-

amplitude loading patterns. The numerical model used the cumulative plastic deformation at the crack tip as the driving force of fatigue crack growth. Through the obtained numerical results, it is possible to draw some more relevant conclusions:

- Considering the same value of minimum and maximum load values, the 15–9 Christmas Tree loading case produced an increase in cumulative plastic strain in comparison with the constant-amplitude loading, particularly during the unloading part of the load block. This effect is more evident without the contact of crack flanks, i.e., without crack closure. The 9–3 Christmas Tree loading case did not show a significant effect, which indicates that the range of small cycles is relevant.
- The effect of the Christmas Tree loading pattern is due to small changes in material hardening. Crack closure is usually used to explain the effect of loading parameters, with the Christmas Tree loading pattern being an exception.
- Damage accumulation was observed after closure, which destroys the purity of the crack closure concept. Crack opening occurred at 21.6 N for the low-frequency load pattern; therefore, the portion of load range during which the crack was closed was 29%. This closure reduced the crack growth rate from 0.33 μm/s, obtained without the contact of crack flanks, to 0.19 μm/s.

**Author Contributions:** Conceptualization, D.M.N. and F.V.A.; methodology, D.M.N.; software, D.M.N.; investigation, F.J. and E.R.S.; data curation, F.J.; writing—original draft preparation, D.M.N.; writing—review and editing, D.M.N., F.V.A. and J.C.N.J.; supervision, F.V.A. All authors have read and agreed to the published version of the manuscript.

**Funding:** The authors gratefully acknowledge the financial support from the Portuguese Foundation of Science and Technology (FCT) under the projects UIDB/00285/2020 and LA/P/0112/2020. Edmundo Sérgio is also grateful to the FCT for the PhD grant with reference 2022.11438.BD.

**Conflicts of Interest:** The authors declare no conflict of interest.

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
