# Peer review of "Fatigue Crack Propagation under Christmas Tree Load Pattern"

_applsci, doi:10.3390/app13031284_

Round 1
Reviewer 1 Report
The paper main focus on evaluate the effect of variable amplitude loadings on fatigue crack. In the paper the so-called Christmas Tree loading pattern is compared with constant amplitude loading patterns, and got some interesting conclusions. The topic of the paper itself has certain theoretical research value.
(1) ‘the mesh size near the crack tip is 8 μm.’ What is the reason for taking 8mm grid? Is there any other relevant reference to demonstrate the grid size? It is recommended to verify and adopt.
(2) In ‘2.3. Load Patterns’, What is the value range of loading force and the basis for the value of frequency? It is recommended to supplement.
(3) The calculation results of the paper lack of relevant experiments or mutual verification with other research results, and it is recommended to supplement.
(4) There are some problems in the drawing, chapter number and writing standardization of the paper, and it is recommended to modify them carefully. For example, Page 10, 351 rows, et al.
Reviewer 2 Report
This paper is well written and the interpretation of the simulation results shows that the authors have expertise in this field. I believe that the manuscript can be published, but before that, a minor revision should be provided. The following points should be considered:
1- In Fig.1, it is suggested to mark where the crack is and show the refined mesh around the crack. Besides, a mesh sensitivity analysis should be performed.
2- In Line 167, the variables a and N in the equation were not explained.
3- In Line 351, a cross-referencing error occurred.
4- There are some style errors in the manuscript, for example, in Line 355, the sentence must begin with a capital letter. Careful check through the main text is required.
5- If possible, please list the conclusions with points supported by quantitative results.
